# Stakeholders’ Perceptions Regarding Adaptation and Implementation of Existing Individual and Environmental Workplace Health Promotion Interventions in Blue-Collar Work Settings

**DOI:** 10.3390/ijerph192013545

**Published:** 2022-10-19

**Authors:** Hanne C. S. Sponselee, Lies ter Beek, Carry M. Renders, Suzan J. W. Robroek, Ingrid H. M. Steenhuis, Willemieke Kroeze

**Affiliations:** 1Department of Health Sciences, Faculty of Sciences, VU University Amsterdam, 1081 HV Amsterdam, The Netherlands; 2Amsterdam Public Health Research Institute, 1081 HV Amsterdam, The Netherlands; 3Department of Public Health, Erasmus University Medical Center, P.O. Box 2040, 3015 GD Rotterdam, The Netherlands; 4Care for Nutrition and Health Group, School of Nursing, Christian University of Applied Sciences, 6717 JS Ede, The Netherlands

**Keywords:** occupational health, workplace health promotion interventions, lifestyle, implementation, MIDI, blue-collar, perceptions, prevention

## Abstract

Blue-collar workers often have disadvantageous health statuses and might therefore benefit from a combination of individual and environmental workplace health promotion interventions. Exploring stakeholders’ perceived facilitators and barriers regarding the combined implementation of these interventions in blue-collar work settings is important for effective implementation. A qualitative study consisting of 20 stakeholder interviews within six types of organisations in The Netherlands was conducted. The potential implementation of the evidence-based individual intervention SMARTsize and the environmental intervention company cafeteria 2.0 was discussed. Data were analysed using thematic analysis with a deductive approach. Five main themes emerged: (1) the availability of resources, (2) professional obligation, (3) expected employee cooperation, (4) the compatibility of the proposed health interventions, and (5) the content of implementation tools and procedures. Generally, stakeholders expressed a sense of professional obligation toward workplace health promotion, mentioning that the current societal focus on health and lifestyle provided the perfect opportunity to implement interventions to promote healthy eating and physical activity. However, they often perceived the high doses of employees’ occupational physical activity as a barrier. We recommend co-creating interventions, implementation tools, and processes by involving stakeholders with different professional backgrounds and by adapting communication strategies at diverse organisational levels.

## 1. Introduction

Studies in European countries point out that blue-collar employees (i.e., those predominantly performing manual labour) often have a low socioeconomic position (SEP) and a related higher risk for health problems such as cardiovascular disease, and early exit from the workforce [1,2,3,4]. Unhealthy lifestyle behaviours and high body weight are more prevalent among these employees than among those with a higher SEP [5,6]. A way to improve these blue-collar employees’ health statuses may be through workplace health promotion (WHP) interventions, which should be oriented toward the individual and the work environment according to the European Network for WHP [7]. The workplace is a suitable setting for implementing such interventions as it enables the reach of large groups of people and the use of existing social connections and support [8,9]. However, little is known about factors specifically associated with successfully implementing the combination of individual and environmental WHP interventions in blue-collar work settings (i.e., non-office settings).

Indeed, the effective implementation of individual and environmental WHP interventions is of the utmost importance. However, combining these interventions is complex and may work best if tailored to the local setting [10]. Prior studies showed that stakeholders’ (e.g., human resource managers and sustainable employment advisors) perceptions regarding such WHP interventions are factors influencing organisational change and implementation, amongst others [11,12]. Thus, they may be vital in influencing the creation of workplace health initiatives [13]. These stakeholders’ perceptions have previously been recommended to be further explored to create a mutual understanding of successful WHP implementation in addition to employees’ perceptions [14]. This exploration should take place before intervention implementation to guide the effective implementation of health promotion initiatives [12]. 

Prior research on stakeholders’ perceptions of factors associated with WHP uptake among employees has been described. A study amongst managers in WHP-related roles showed that they mainly perceived barriers regarding organisational support (e.g., lack of time and training for managers) required for effective WHP implementation [15]. Another study on stakeholders’ experiences after implementing workplace health and wellbeing initiatives also mentioned a lack of time as a barrier to staff engagement [16]. These studies have mainly focused on individual instead of environmental WHP interventions, though the physical and social work environment plays a role in influencing individual employees’ health behaviours, such as eating behaviour and physical activity [17,18]. A prior study focused on the perceptions of combined (i.e., individual and environmental interventions) implementation and found that contextual factors such as workplace culture and organisational structure must be considered [12]. However, these aforementioned studies were conducted after intervention design and implementation, not specifically focused on blue-collar work settings.

This study aims to explore perceptions of stakeholders (i.e., those related to intervention implementation) in The Netherlands regarding the future implementation of the combination of two evidence-based WHP interventions. To the best of our knowledge, this is the first study to explore perceptions of such a diverse range of stakeholders (i.e., from those responsible for implementation at the executive to strategic levels), to find directions to tailor interventions to specifically blue-collar work settings. The individual WHP is lifestyle intervention SMARTsize (i.e., focused on healthy eating behaviour, physical activity, and weight management) [19,20] and the other is the environmental intervention Company cafeteria 2.0 (i.e., focused on a healthy food environment to facilitate healthy eating behaviour) [21]. Furthermore, stakeholders’ perceptions regarding required tools that facilitate effective and sustainable implementation are explored. For this study, stakeholders are defined as those with responsibilities in policy and management positions in their organisations, influencing the potential implementation of WHP interventions. Characteristics of the intervention, adopting person, organisation, and socio-political context will be mapped based on the measurement instrument for determinants of innovations (MIDI) [22], complemented by characteristics of the implementation strategy. This study is embedded in the intervention design phase of the larger research project SMARTsize@Work, aimed at improving blue-collar employees’ eating behaviour and physical activity in The Netherlands.

## 2. Materials and Methods

### 2.1. Design, Study Population, and Recruitment Procedures

This study had a qualitative design in which 20 individual semi-structured interviews with stakeholders within blue-collar work settings in The Netherlands were held.

Participants were recruited in cooperation with one of the largest catering companies in The Netherlands. This collaboration was chosen to facilitate the recruitment of relevant stakeholders, as this company provides catering for clients (i.e., organisations) whose workforce includes large numbers of blue-collar employees. The research team and the catering company agreed that the catering company’s role in recruitment was to initiate contact with relevant organisations from their database. Subsequently, the first author (HS) contacted organisations to further discuss the participation of relevant stakeholders. Stakeholders’ inclusion criteria were having an organisational or managerial position related to the potential implementation of health interventions and being employed in an industry with a substantial number of blue-collar employees (e.g., the logistics, hospitality, and construction industries). Moreover, industry selection was based on a report on the labour market position of employees with low and medium levels of education in The Netherlands [23].

Contact persons from 27 blue-collar organisations were approached through the catering company’s internal communication channels, accompanied by an e-mail and flyer provided by the research team. Reasons for not participating were time constraints and reorganisations. A total of 20 stakeholders from the following six different types of organisations were willing to participate in individual interviews: an air transport organisation (*n* = 2), an automotive and consumer goods organisation (*n* = 3), an organisation focused on food services and facilities management (i.e., the catering company that cooperated in this study’s recruitment; *n* = 5), a governmental organisation (*n* = 2), a motor vehicle manufacturing organisation (*n* = 6), and a pharmaceutical production organisation (*n* = 2). In addition, multiple individual interviews per organisation were planned with stakeholders with different organisational positions to comprehensively explore stakeholders’ perceptions. Table 1 gives an overview of the organisational characteristics related to the participants, sorted alphabetically by organisational industry and department of position. 

### 2.2. Data Collection and Procedures

The first author (HS) held the interviews at the participants’ workplaces. All participants gave written informed consent. Prior to the interviews, the first author explained the background of the study and the existing individual and environmental WHP interventions to be discussed. The individual intervention was SMARTsize, a lifestyle intervention aiming to promote weight management and sustainable changes in healthier eating behaviours through an emphasis on portion size and calorie density [19], which was recently further developed by adding a physical activity component [20]. The intervention consisted of a website with interactive quizzes, a tool to identify the home eating environment (i.e., the Homescreener), cooking workshops, and individual consultations with a healthcare professional. The first author showed participants SMARTsize materials to support the intervention explanation: a screenshot of the website, the Homescreener, a worksheet about making a meal less calorie dense and one about eating habits used during individual consultations. The environmental intervention was the company cafeteria 2.0, which aims to promote healthy food purchasing and consumption through nudging and social marketing techniques [21]. The intervention consisted of 14 strategies to create a healthier company cafeteria, involving product, price, placement, and promotion. To support the intervention explanation, the first author showed participants the practical guide describing these strategies intended for stakeholders (i.e., facility managers) who want to create a healthier company cafeteria [24].

The interview guide was developed based on the MIDI, a diagnostic tool to gain insight into 29 determinants (facilitating and obstructive factors) for using innovations in healthcare [22]. The MIDI was used because it is a systematic way to explore implementation processes of innovations. The MIDI’s determinants are divided into four categories of the innovation framework: (1) the innovation, (2) the adopting person, (3) the organisation, and (4) the socio-political context. In this study, a fifth category regarding characteristics of (5) the implementation strategy was added to explore implementation approaches specifically. Of the 29 determinants described in the MIDI, 17 were selected based on their applicability for adapting existing interventions to a new setting. Determinants related to existing innovations were omitted as we explored perceptions prior to implementation. 

The MIDI’s determinants and associated operationalisations formulated into statements were rephrased into questions corresponding with the purpose of this study to explore stakeholders’ perceptions prior to implementation instead of inquiring about their experiences with existing interventions. For example, the determinant ‘Compatibility’ described the operationalisation of ‘The innovation is a good match for how I am used to working’, which was rephrased to ‘To what extent do you think the intervention suits the way you are used to working?’ The word ‘innovation’ was replaced by the word ‘intervention’ due to the aim of this paper. The main topics of the final interview guide and sample questions can be found in Appendix A.

The interviews lasted for a minimum of 33 and a maximum of 54 min and were recorded (Olympus WS-853) and securely stored. The interviews were conducted between May and July 2019.

### 2.3. Data Analysis

Interviews were transcribed verbatim. Transcriptions were anonymised and subsequently imported and analysed in the MAXQDA software (2020, VERBI Software: Berlin, Germany) package for qualitative data analysis. The data were analysed using thematic analysis [25] with a deductive approach using a broad framework for the coding process [26,27]. First, the main categories for coding were added regarding both proposed health interventions (i.e., the lifestyle and healthy food environment intervention). Then, the deductive approach was used by adding the four categories of the MIDI (characteristics of the innovation, the adopting person, the organisation, and the socio-political context) and the added category of the implementation strategy as subcategories under the main categories of the two interventions. Each subcategory was then supplemented with the two categories of facilitators and barriers, in line with the research question. 

Before the analysis, two authors (HS and LB) familiarised themselves with the transcripts. Then, they independently generated initial codes for five interviews within the categories as described above. Subsequently, they discussed differences in a consensus meeting, which led to developing a coding tree. Next, four authors (HS, LB, WK, and CR) discussed this coding tree and agreed on the final codes, and then the lead author (HS) coded the remaining 15 interviews. Finally, the second encoder checked these coded transcripts and discussed and agreed on the final codes with the lead author. Both researchers kept a logbook during all coding phases to reflect on their perspectives when thematising the data. The data were originally collected in Dutch. The quotes were translated into English after data analysis. 

## 3. Results

Based on the data analysis, five main themes regarding the future implementation of individual and environmental WHP interventions emerged: (1) the availability of resources, (2) professional obligation, (3) expected employee cooperation, (4) the compatibility of the proposed health interventions, and (5) the content of implementation tools and procedures (Table 2). Each theme emerged within a predetermined category. No distinct theme emerged within the ‘Characteristics of the socio-political context’ category. This category is operationalised by one determinant describing existing legislation and regulations by competent national authorities [22], which was not represented by the data.

No distinct themes emerged within the category ‘Characteristics of the socio-political context’. Each theme is detailed below, covering both interventions and describing associated facilitators and barriers. The interconnectedness between themes is described when relevant.

### 3.1. The Availability of Resources

Stakeholders mentioned barriers regarding resources for both interventions. Some stakeholders mentioned that it would be difficult to execute the lifestyle intervention at the workplace and during working hours due to a lack of time for such activities due to employees’ short breaks and their tasks being directly related to the organisation’s profits. For example, a human resources manager at a motor vehicle manufacturing organisation mentioned the following:


*A lot of practically trained staff […] have a direct relationship with the end product [i.e., trucks] we make. They are therefore clocked. They are therefore monitored in terms of time span, how much they are working and that ultimately determines the cost price of a truck.*


Stakeholders mentioned that certain working conditions (e.g., high doses of occupational physical activity) and a variety of working hours would negatively affect implementation among these employees. They often work shifts at irregular working hours, which might hinder consistent intervention organisation and group participation. Regarding the healthy food environment intervention, material resources and facilities were often perceived as inadequate. Examples were limited equipment (e.g., a lack of a refrigerator at the counter to sell pre-cut fruits), small sales space, and a disadvantageous logistically set-up restaurant that could not be rebuilt. A catering manager at a pharmaceutical organisation mentioned,


*But here, the fact remains that they [i.e., employees] pass by the warm food first because that is where the entrance is. That is where they pick up their plate, so that is where they pass by the snacks first.*


A few stakeholders also mentioned the formal ratification as a barrier to food environment intervention. For example, some organisations had contracts with caterers without having the opportunity to adapt to the portion sizes provided at their worksite cafeterias. Thus, stakeholders mentioned barriers to implementation related to available resources for both the lifestyle and healthy food environment interventions.

### 3.2. Professional Obligation

Stakeholders were asked to what extent they experienced the responsibility of offering employees individual lifestyle and healthy food environment interventions. They mentioned a degree of professional obligation towards offering their employees a lifestyle intervention or contributing to improving their lifestyles, including facilitating a healthy food environment at the workplace. For example, a member of a working council at an automotive and consumer goods organisation mentioned the following: 


*What matters to me is that employees are given the opportunity to participate and that it is encouraged by the employer. I think that is important. I do not mean that the employee or the employer is obliged to do it, not at all, but better offers help us all.*


Stakeholders often explained that both proposed interventions aligned with their organisational policy. Indeed, many stakeholders indicated that their organisation had a personnel policy aimed at sustainable employability, addressing lifestyle and other health-related factors (e.g., stress and debt management). In addition, several stakeholders indicated that they had already focused on employees’ health but were still in the process of setting up the policy.

A line of thought regarding professional obligation towards facilitating a healthy food environment at the workplace was the sense that the responsibility for healthy living should be shared between the employer and employee. Furthermore, many stakeholders expressed that they doubted what the employer’s role in employee health is or should be. To conclude, most stakeholders expressed a sense of obligation towards WHP activities. However, the extent to which they are responsible for employees’ health remained an important question.

### 3.3. Expected Cooperation of Employees

Stakeholders were asked to what extent they thought the envisioned end-users of blue-collar employees would cooperate in the proposed health interventions. Many stakeholders reacted that the proposed interventions would be relevant for the employees, but they expected employee participation to be challenging. The underlying reasons were different for the two interventions. Regarding the individual lifestyle intervention, stakeholders reasoned it would be difficult to implement because employees would distrust the employer’s use of their health information when they required sick leave. A human resources manager at a motor vehicle manufacturing organisation mentioned,


*I also know that many people have not taken the fitness test because they are afraid that the information it provides will be passed on to the employer, that the employer will do something with it. […] That brings us to the point of interference, while in my opinion, or in my view, I know that it’s not even allowed under the privacy legislation, but people are still wary of it, that it will be used or misused. Well, that’s one side of the coin: ‘OK, you’re offering it to me, but why? What’s in it for you? So are you going to misuse this now if it turns out that I am very unhealthy?’*


Another barrier to the lifestyle intervention was the high doses of occupational physical activity among blue-collar employees. Stakeholders often mentioned that occupational physical activity caused these employees to usually be sufficiently physically active during working hours, which led them to expect less cooperation among these employees in lifestyle interventions, including a physical activity component. For example, a human resources advisor at an automotive and consumer goods organisation mentioned,


*Especially in our production, everything is standing up, and people are also walking a lot, so I can imagine. I recently spent a day in production. I was glad to be able to sit down. So, well, walking around during lunchtime is a little less attractive, I think, for a production worker, or at least here.*


Reasons regarding less expected cooperation for the healthy food environment intervention were often related to the suitability of the intervention for blue-collar employees, for example, because these employees often bring their own food instead of eating at the worksite’s cafeteria. Moreover, healthy food was expected to be too expensive. Additionally, the stakeholders suggested that promoting healthier alternatives in the food environment might lead employees to make unfavourable food choices because they might disagree with the changes. Overall, stakeholders linked the expected cooperation to the compatibility of the health interventions, emphasising the employees’ potential mistrust and high occupational physical activity levels.

### 3.4. Compatibility of the Proposed Health Interventions

The extent to which stakeholders perceived the health promotion interventions as compatible with how they themselves were used to working was discussed. Many stakeholders mentioned that both health-promoting interventions were overall compatible with their organisational policy and mission regarding employees’ health. However, they highlighted certain downsides regarding the content of the proposed interventions, often related to their perceived barrier to employees’ cooperation. Regarding the lifestyle intervention, some stakeholders missed a focus on sustainable employability or ergonomic occupational physical activity, especially as this group has a high occupational physical load. Adding the component of sleep behaviour as part of the intervention was also mentioned because this group often deals with shift work in which working hours may vary greatly.

Regarding the proposed healthy food environment intervention, it was often mentioned that it was too complex. Many stakeholders reasoned it was too challenging, for example, to adapt proposed portion sizes in the intervention due to the present company culture, which was not receptive to smaller meals or logistics such as food being delivered in predetermined packaging. In line with this compatibility, it was also mentioned that the current societal focus on health and lifestyle offered the perfect momentum for implementing such interventions, which might be linked to their perceived professional obligation. An account manager at a food services and facilities organisation explained,


*I think the society at the moment is just peppered with this topic.*


To conclude, both health interventions were generally perceived as compatible with the stakeholders’ way of working, though compatibility with the employees’ working conditions and company culture might hinder implementation and employee cooperation.

### 3.5. Content of Implementation Tools and Procedures

Stakeholders were asked what would help them offer both interventions. They emphasised that communication about the intervention should be tailored to all organisational levels (i.e., implementers such as HR managers, catering managers, direct team leaders, and blue-collar employees). These levels should all be involved to achieve support and successful implementation. Regarding the content of implementation tools, it was emphasised for both interventions that the goal of the intervention should be clear to the implementers (e.g., managers and team leaders) and employees. 

Concerning implementers, implementation could be facilitated when the goal of sick leave reduction due to intervention implementation is communicated. Towards employees, it seemed important not to emphasise ‘healthy’ because that would imply imposing on employees how they should behave. Regarding the intervention delivery, it was often mentioned that implementation should be carried out easily and that the intervention should be delivered ready-to-use. A few stakeholders mentioned that they would perceive the implementation as a hindrance if it was too burdensome. A catering manager at a motor vehicle manufacturing organisation mentioned,


*[…] as long as it’s not too much work, so to speak. It doesn’t have to be such a big deal to have that effect.*


Furthermore, barriers and facilitators regarding implementation channels and procedures were mentioned. For employee participation, digital communication tools regarding recruitment and implementation were expected to be a barrier because of this group’s job type and working conditions. They often do not have work-related access to the internet because they do not use laptops or company e-mail addresses. Regarding procedures, two stakeholders emphasised that the change should be made incrementally. A few stakeholders also mentioned that implementing this intervention should just be started and tried out, as it is a first step that allows employees to slowly become used to the intervention offered. A human resources manager at a governmental organisation offered,


*So, I do believe that sometimes you just have to do it. Of course, you can ask people all day long: ‘What do you want? And how can we reach you?’ But sometimes, I think that as an organisation, you just have to offer something to help people, but you also have to simply make decisions you want to communicate as an organisation.*


Thus, stakeholders mentioned various aspects of implementation tools (e.g., tailored goal-oriented communication) and procedures (e.g., changing incrementally) that would support them in offering the proposed health interventions.

## 4. Discussion

Five main themes emerged related to exploring stakeholders’ perceptions regarding facilitators and barriers to implementing the combination of an existing individual and environmental WHP intervention for blue-collar workers: (1) the availability of resources, (2) professional obligation, (3) expected employee cooperation, (4) the compatibility of the proposed health interventions, and (5) the content of implementation tools and procedures. Generally, stakeholders’ perceptions varied for the proposed individual and environmental interventions. Facilitators and barriers of both health promotion interventions have been described when present. Below, the results are discussed according to the main themes, emphasising results that seem specific to the blue-collar work setting.

### 4.1. Discussion of Main Themes

#### 4.1.1. The Availability of Resources

Stakeholders mentioned various limiting factors for implementation related to resources for both interventions. Barriers to the lifestyle intervention were mainly linked to resources, such as a shortage of employees’ time. This finding may be specifically related to the blue-collar work setting, as often a direct relationship exists between physical working hours and company profits. In comparison, white-collar jobs are often more oriented towards management and information processing tasks, which tend to be related to having more freedom to flexibly manage work tasks. Interestingly, a shortage of the stakeholders’ time amid other responsibilities was not mentioned, although it was found in prior research [15,16]. Therefore, our finding on a lack of perceived employee time may be specifically considered relevant in blue-collar work settings. 

The direct relationship between employees’ time and organisational profits was mentioned, although stakeholders did not explicitly mention a lack of financial resources to cover possible implementation costs. Other studies have shown that financial constraints might be a barrier to implementation [16] and that removing financial barriers might facilitate employer WHP implementation [13]. A systematic review indicated that WHP activities could contribute to positive changes in weight-related outcomes of employees [28], and improvement of weight status might result in less absenteeism due to illness and overall impairment at work [5,29]. This improvement requires a long-term view on employability, which an increasing number of organisations seem to possess [30]. However, the perceived limited resource of employees’ time in the short term might hinder this long-term vision, which is necessary to implement preventive health promotion activities.

#### 4.1.2. Professional Obligation

Generally, stakeholders expressed a sense of obligation towards WHP activities. Similarly, another study found that employers and other stakeholders (e.g., labour unions and insurance companies) perceived responsibility in connection with their professional obligation [31]. However, it was also described that employers often do not feel responsible for their employees’ health behaviours unless there is a risk of injury [31,32]. This way of stakeholders perceiving responsibility might differ depending on the organisations’ focus on health initiatives [16] or the stakeholders’ organisational position.

Furthermore, doubt about the employer’s versus employee’s role in employee health emerged in many interviews. This doubt endorsed the fundamental question: Who is actually responsible for employee health? Indeed, questions such as how far an employer can go in promoting employees’ health have often arisen [33]. In The Netherlands, occupational healthcare operates in a private market and strongly depends on the contract between an occupational health and safety (OHS) service and the organisation. OHSs primarily focus on work-related issues, which may be a barrier to implementing preventive interventions focusing on life domains indirectly associated with work, such as eating behaviour and physical activity [34]. Presumably, the type of contract between the OHS and the organisation might be related to the stakeholders’ sense of obligation and perceived responsibility for employee health. 

#### 4.1.3. Expected Cooperation of Employees

Stakeholders expected cooperation to be challenging, although they perceived the proposed interventions relevant for blue-collar employees. Another study likewise found that stakeholders perceived employees with a lower SEP to be hard to reach for participation in preventive interventions [34].

A key reason for expected low cooperation in the lifestyle intervention was the high amount of occupational physical activity among blue-collar employees. In many jobs, occupational physical activity is ill-proportioned, causing negative health outcomes (e.g., cardiometabolic and musculoskeletal problems) instead of promoting health [35,36]. This result might prevent lifestyle intervention participation by employees. However, our other study in the SMARTsize@Work project showed that blue-collar employees were quite interested in participating in the lifestyle (i.e., eating behaviour and physical activity) component of a WHP programme [14]. 

A similar gap between health care professionals’ perceptions and patients’ experienced realities has been previously described in obesity research [37]. A reason for the expected low cooperation for the healthy food environment intervention was that it would not be perceived as suitable because these employees bring their own food instead of eating at the worksite’s cafeteria, and the food there would be too expensive. Therefore, it might be interpreted that a gap exists between a more pessimistically perceived employee willingness and receptiveness by stakeholders and actual employee willingness.

#### 4.1.4. Compatibility of the Proposed Health Interventions

Stakeholders generally perceived both interventions as compatible with the current societal focus on health and lifestyle behaviour and with the organisation’s personnel policy, which often focused on sustainable employability. However, the lifestyle intervention was mentioned to miss an ergonomic occupational physical activity component for these employees with high occupational physical activity levels. Moreover, the need for sustainable employability interventions for employees with a low level of education was previously found [38,39], emphasising that focusing on work conditions should be part of an integrated approach (i.e., integrating employees’ health and occupational safety) when aiming at improving employees’ health [40].

Furthermore, the healthy food environment intervention was mentioned as not always being compatible with the company culture where employees would not be receptive, for example, to smaller meals. A previous study also concluded that employees with an average lower level of education were less receptive to making healthy food choices in an environmental WHP intervention because health was found to be less concerned in this group [41]. This idea might illustrate a more fundamental misconception of both stakeholders and employees that ‘healthy’ foods do not meet the needs of employees with high doses of occupational physical activity.

#### 4.1.5. Content of Implementation Tools and Procedures

Generally, stakeholders mentioned that the interventions’ communication should be tailored to stakeholders at different organisational levels (i.e., implementers such as HR managers, catering managers, direct team leaders, and blue-collar employees), emphasising the end goals of participation and implementation. These findings are consistent with other research on stakeholders’ perceptions, suggesting that effective communication is a key facilitator of implementation [11,16] and that strong collaborations between stakeholders may be crucial to achieving successful implementation through organisational support [12]. Implementation tools and procedures appear to require effective communication to all organisational levels involved. Previous research on potential underlying mechanisms of effective health communication, emphasizes that tailoring the message, source and channel to intended audiences is crucial [42]. From an individual perspective, this finding is in line with previous research on human information processing. The Elaboration Likelihood Model describes the way people process information and change attitudes [43], and a tailored communication approach might be effective because people tend to pay more attention to information they consider personally relevant [44]. From a broader perspective, this tailored individual communication should fit the system it is part of. The Diffusion of Innovations Theory [45] served as one of the foundations of developing the MIDI [22], and describes how communication about innovations spreads through a social system over time. The structure of this social system determines to what extent individuals affect each other. Therefore, the individual tailored communication of the implementation of WHP interventions (i.e., the innovation) should fit the broader blue-collar work setting (i.e., the social system).

### 4.2. Direct and Indirect Barriers Regarding Implementation in Blue-Collar Work Settings

Both direct and indirect barriers to the implementation of the lifestyle intervention (i.e., focused on eating behaviour and physical activity) in blue-collar work environments will be emphasised here, recurring throughout the main themes. The direct barrier was related to the actual nature of blue-collar jobs, which are often already high in occupational physical activity and often have a more direct link to organisations’ profits. This limits the available time and the potential organisational willingness to implement WHP interventions. A lack of employees’ time for these activities might act as a fundamental resource barrier specifically perceived by stakeholders in blue-collar work settings.

The indirect barrier was related to the expected cooperation of blue-collar employees due to their high doses of occupational physical activity. Stakeholders perceived these employees not to be interested in a physical activity component as part of the individual intervention. Additionally, they perceived cooperation barriers regarding healthier eating behaviour as these employees with high levels of physical activity were perceived as not being receptive to consuming food in, for example, smaller portions. Stakeholders might have thought that the proposed interventions were insufficiently adaptable to their specific organisational and employees’ needs, although they could be tailored to the preferred circumstances.

These perceived barriers might discourage stakeholders from providing employee WHP interventions at all, possibly widening the gap between stakeholders’ and employees’ perceptions of employees’ willingness to participate even more. Therefore, it might be interpreted that a gap exists between a more pessimistic perceived employees’ willingness and receptiveness by stakeholders and true employees’ willingness.

### 4.3. Strengths and Limitations

This study contributes to a better understanding of stakeholders’ perceptions regarding facilitation of individual and environmental worksite health-promoting interventions, and barriers related to characteristics of the blue-collar work setting. The findings add to the literature by describing how stakeholders’ perceived barriers, of which some seem specific for blue-collar work settings, might hinder WHP provision to employees in the first place. Using the MIDI [22] as a theoretical framework to support this exploration is a strength because it systematically guided the interviews and analysis. In addition, although the MIDI was developed initially for usage in healthcare instead of work settings, it provides a generic description of determinants that can be applied in other contexts [46]. To our knowledge, this was the first study that used the MIDI to guide the process of finding directions according to stakeholders to adapt evidence-based individual and environmental WHP interventions to blue-collar work settings.

A limitation might involve a sampling bias, as the stakeholders who agreed to participate might have had a more than mainstream interest in employee health and associated interventions. Recruitment was facilitated by approaching the clientele of a large catering company. However, we included various organisations and believe we would have found similar main themes by recruiting stakeholders in other organisational industries through different recruitment strategies. Moreover, new themes might have emerged when exploring stakeholders’ perceptions of different professional positions, organisations, or areas in The Netherlands. Therefore, data saturation was difficult to justify. However, the findings suggested satisfactory data saturation because the 19th and 20th interviews did not reveal new themes.

### 4.4. Implications for Practice and Research

We recommend that the perceptions of stakeholders (i.e., those responsible for intervention implementation) in blue-collar work settings should be included in the implementation of health interventions, which can be achieved by considering their perceptions during the development of interventions, implementation tools and implementing interventions, in line with previous recommendations [12]. Additionally, these stakeholders’ perceptions should be complemented with blue-collar employees’ perceptions to create mutual understanding, which we have explored previously [14]. We have explored stakeholders’ perceptions separately because they deal with different organisational considerations regarding WHP implementation than the end-users of employees. In line with previous recommendations [31], we recommend co-creation as a next step to involve both end-users and stakeholders from different organisational levels in the design and implementation of such health interventions. It is important to bring in scientific knowledge on WHP interventions during this co-creation process, to combine experiences from practice with science. This can help ensure that WHP interventions are properly tailored to specific settings, such as blue-collar work settings. Health promotion activities in white-collar work settings might often not be transferable, often focusing on different health behaviours, such as the reduction in prolonged sitting hours [47]. As both work settings differ based on, for example, physical employee productiveness related to the organisation’s profits, these disparities must be specifically considered when developing and implementing health promotion interventions to contribute to closing the occupational health gap.

An implementation effort for blue-collar work settings in line with previous recommendations might be to think outside the box regarding health promotion activities [48] by offering individual health interventions partly during and after working hours [14]. This approach might compromise the stakeholders’ perceived barriers to employees’ participation during working hours and simultaneously enable more inclusive participation appropriate for a wider range of working conditions (i.e., including those with shift work at irregular hours). The healthy food environment intervention within the blue-collar work setting can, for example, apply nudging strategies outside the worksite cafeteria when these employees do not eat there. For example, increasing the availability of healthier items in vending machines may increase sales [49], or introducing a fruit basket might increase the availability of healthy food in the work environment [50,51]. Furthermore, stakeholders should be provided with suitable implementation tools, including knowledge on selecting and implementing these WHP interventions [52,53]. They should also be provided with explicit information emphasising the flexibility of implementation strategies to facilitate the adaptation of intervention implementation in their work setting.

Future research could explore the specific implementation tools, including effective communication strategies, that should be designed for implementing combined individual and environmental health interventions in blue-collar work settings (e.g., handbooks, guidelines). Train-the-trainer programmes for stakeholders involved in implementing health interventions in blue-collar work settings might be a starting point to help with using these implementation tools correctly and effectively [54], appropriate to the specific context of the work setting. Subsequently, a process evaluation should investigate whether these tools facilitate stakeholders to implement individual and environmental WHP interventions in blue-collar work settings.

## 5. Conclusions

Stakeholders had a sense of professional obligation toward WHP in blue-collar work settings. However, it was doubted whether employers or employees should mainly take responsibility for employee health. The proposed individual and environmental health promotion interventions were mentioned to be compatible with their organisation’s human resource policies and with the current social focus on health. High doses of employees’ occupational physical activity, a shortage of employees’ time, and expected low cooperation were identified as key barriers. These barriers might hinder stakeholders from offering WHP interventions in the first place, limiting employees’ chances to participate. Co-creating WHP interventions and implementation procedures together with multiple organisational levels (i.e., stakeholders responsible for WHP implementation and the end-users of blue-collar employees) might increase intervention uptake and contribute to their effectiveness.

## Figures and Tables

**Table 1 ijerph-19-13545-t001:** Organisational characteristics related to the study participants (*n* = 20).

Organisational Industry	Participant’s Organisational Department of Position	Organisational Position
Air transport		
	Human resources	Manager
	Human resources	Project Manager
Automotive and consumer goods		
	Human resources	Advisor
	Human resources	Manager
	Works council	Member
Food services and facilities management		
	Account management	Account Manager
	Food Safety	Manager
	Marketing	Manager
	Quality, Health, Safety, Environmental	Manager
	Site Services	Manager
Governmental		
	Account management	Contract Manager
	Human resources	Manager
Motor vehicle manufacturing		
	Catering	General Manager
	Catering and facilities	Manager
	Human resources	Programme Manager
	Human resources	Manager
	Works council	Member
	Works council	Member
Pharmaceutical		
	Catering	General manager
	Catering	On-site manager

**Table 2 ijerph-19-13545-t002:** Themes resulting from the data analysis, categorised by facilitators and barriers of both health promotion interventions.

	Predetermined Category		Individual Health Promotion Intervention	Environmental Health Promotion Intervention
The availability of resources	Characteristics of the organisation (MIDI)	Facilitators	x	x
		Barriers	Employees’ time, related to short breaks and job tasks directly related to the organisation’s revenue Employees’ variable working conditions and working hours that hinder the organisation of the intervention and group participation	Hindrance of implementation due to material resources and facilities such as: - *Limited equipment* - *Small sales space* - *Disadvantageous kitchen* *logistics*
Professional obligation	Characteristics of the adopting person (MIDI)	Facilitators	Perceiving a degree of professional obligation regarding: - *Contributing to employee health*	Perceiving a degree of professional obligation regarding: - *Providing a healthy food environment for employees*
		Barriers	x	Formal ratification, such as a contract with a caterer that interferes with changes in the company cafeteria Promoting healthier alternatives could lead employees to make unfavourable food choices Responsibility for employee health should be shared with employees
Expected cooperation of employees	Characteristics of the adopting person (MIDI)	Facilitators	x	x
		Barriers	Low expected employee cooperation due to: - *Distrust in the employer by the employees* - *High doses of occupational physical activity, hindering interest in participating in the physical activity component of the intervention*	Low expected employee cooperation because: - *These employees often bring their own lunch instead of eating* *in* *the company’s cafeteria* - *Healthy food* *would* *be too expensive for these employees*
Compatibility of the proposed health interventions	Characteristics of the innovation(MIDI)	Facilitators	Overall compatibility with the organisation’s employee health policy and mission Current societal focus on health and lifestyle	Overall compatibility with the organisation’s employee health policy and mission Current societal focus on health and lifestyle
		Barriers	Intervention lacks a focus on: - *Sustainable employability or ergonomic occupational physical activity for this group with high* *dose* *occupational physical activity* - *Improving sleep behaviour* *of* *employees with shift work*	The complexity is too high in terms of: - *Adjusting* *portion sizes in the company cafeteria* - *The work* *culture that is not receptive* - *Foods that are* *often delivered in predetermined packaging*
			**Individual and environmental health promotion intervention**	
Implementation tools and procedures	Characteristics of the implementation strategy(added category)	Facilitators	Tailor the communication of the interventions to all organisational levels (i.e., implementers such as HR managers, catering managers, direct team leaders, and end-users) Involve all organisational levels in the implementation process to achieve support and successful implementation Make sure the goal of the implementation is clear to implementers and employees Take into account the easy execution of the implementation Ensure that the intervention is delivered ready-made Introduce the changes implied by the intervention step by step Try the implementation out to let employees slowly become used to the interventions	
		Barriers	Implementation is too much hassle Reaching employees with digital communication tools because their jobs are not performed digitally	

## Data Availability

The datasets generated and analysed during the current study are not publicly available because study participants did not explicitly agree that their raw data would be shared publicly but are available from the corresponding author upon reasonable request.

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
