# Peer review of "Stakeholders’ Perceptions Regarding Adaptation and Implementation of Existing Individual and Environmental Workplace Health Promotion Interventions in Blue-Collar Work Settings"

_ijerph, 2022, doi:10.3390/ijerph192013545_

Round 1
Reviewer 1 Report
In general, this is an interesting study as research with blue color workers is rather rare. Doing a quali study is fine and conclusion for co-creation is timely. However, I really miss and accordingly require that the authors
1) include more theoretical work i.e. theories and models to build your research on and to integrate the findings in within the discussion section. Such a theory could be the JDRM, but any other theory is also fine.
2) analyze and aggregate your data by means of identifying categories and concepts. Your Table 2 and the text are fine, but the aggregation and synthesis needs to be added, too. A figure would be favored, too, because that does not only helps you with this needed step but also to increase readability and comprehension of your paper.
3) your discussion is rather lengthy and needs to become more structured and more relating to other work. Please include the theory you chose relating to my point 1 above by meeting this point and improving your discussion. This is really important not only from a scientific point of view but also in terms of suggestion tools to the stakeholders on basis of your results.
4) tease out more from their data regarding co-creating because in the abstract you put this up from very prominently, but then there is nothing in the results section and rather little in the discussion. This is very sad as the reader expects more on this topic, also because much research is going on in this field currently, especially on the EU-level.
Reviewer 2 Report
Congratulations to a nice paper and thank you for the opportunity to read this paper! In my opinion, it is a paper that is nicely written with a good structure and readability. It concerns an important topic that adds valuable information to the field of implementation of WHP intervention. In the following, I will give some comments and suggestions mostly regarding aim and methods.
Method
When reading the title, abstract and introduction I understand the paper as it is exploring implementation of individual and environmental WHP-interventions in general. However, when reading the data collection and procedures (line 119-136) it seems to be two particular interventions that is being discussed and explored. If that is correct this needs to be clearly stated in the title and the aim, not to mislead the readers.
Regarding the analysis it is a bit difficult to really understand how you used the categories from MIDI in the analysis procedure. Were codes initially sorted into these categories or was the data first divided into the categories and thereafter coded? Please expand the description of the analysis procedure.
As the procedure is not totally clear and the results presented five main themes that are labelled independently from the MIDI one starts to wonder whether the analysis is deductive or if it is an abductive approach that is being used. Please elaborate on this and explain in what way the analysis is deductive (if you still think so).
In what phase was the data translated? Please, add some information about that.
Discussion
The discussion is interesting, relevant and well-written. I just have a minor comment regarding line 340 where you write that all themes included facilitators and barriers. I might misunderstand table 2 but to me it look like there are some themes without any facilitators identified (eg. The availability of resources). Please check this and clarify.
Reviewer 3 Report
The authors report on an interview based study conducted with 20 managers from 6 organizations in the Netherlands to obtain their perspectives and perceptions regarding the implementation of workplace health promotion intervention called SmartSize that would specifically focus on blue collared workers in their respective organizations. The study sample was largely from the catering domain. The study used a thematic analysis with a deductive approach to identify five key themes.
Overall the study topic regarding workplace health promotion and blue collared work settings is an important one. For the most part, the study was adequately described. I do have some comments and concerns related the article which I have described in my feedback below. I hope the authors find the feedback to be constructive and useful.
General Comments
One key concern is that the specific novel/new scientific contributions of this study were not clearly stated. The methodology and findings were easy to follow. However, the novelty of the methods or new contributions of the findings in relation to what is already known in the literature about workplace health interventions needs to be clarified.
The Introduction section could use some additional context for the study. For instance, how is “blue-collared workers” and “blue-collared work settings” defined for the purposes of this study? Are the poor health outcomes about blue-collared workers mentioned in the Introduction discussing specific occupational domains or all domains? Developed countries or LMICs, or both? If the trends span different domains, and/or types of countries then this must be mentioned in the introduction, else it reads to be very broad and vague. Likewise the type and effectiveness of interventions mentioned in the Introduction could use some description and examples. Providing some specificity to these key concepts would only strengthen the article.
It was only in line 87 of the study aims did it become clear that this study and sample (N = 20) was specific to the Netherlands. It further became clear in the Methods that these were workers specifically from the catering domain. These aspects should be mentioned in the Abstract, and also earlier in the Introduction when explaining blue-collared work settings (see comment 1 above) and in the context of the Netherlands. The generalizability of the findings to other countries and blue-collared work settings could be explained in the Discussion section – but the Title, Abstract, Introduction and Aims must be specific to the study scope, else it appears misleading.
Only in the Methods section did it become clear that the interviews conducted was regarding a specific intervention called SmartSize, and not just “embedded in the intervention design phase of the larger research project SMARTsize@Work” (line 85) as stated in the Aims. I found it somewhat misleading. If the intent of the interviews was the gather information about the current or new features of an existing intervention (SmartSize), then that must be at least be mentioned in the Introduction and be part of the Aims, and if possible also mentioned in the Abstract. Please revise.
This article focuses on “stakeholder perspectives”. However the introduction only describes stakeholders as “Stakeholders’ (e.g. human resource managers and sustainable employment advisors)” (lines 53-54). A more detailed description of who is considered as stakeholders and why, for the purposes of this article (scope of the study) is needed. Nonstandard terms such as sustainable employment advisors need to be discussed with some examples, e.g. I don’t know if these advisors are common in LMICs. Could the study sample be considered as “white-collared workers”? Kindly clarify these aspects in the article.
The interview questions were developed by adapting items from the “measurement instrument for determinants of innovations (MIDI)” (line 83). While I am not questioning the choice, it would be helpful if the authors also explained in the Methods why they chose MIDI. Are there other frameworks or instruments in the literature that the authors considered but decided against, and if so then why (e.g., advantages and limitations of MIDI relative to other options)? This would be useful to also explain in the Methods or possibly in the Discussion section.
Since the interview questions was not a standard questionnaire and was adapted from other sources to suit the study, it would be helpful if the questions were provided verbatim in the Appendix. This would help to better understand the context, and interpretation of the responses and findings.
It was not clear why the lack of worker perspectives in the study was not included as a limitation. Rather it was discussed under Implications. I say this because, in terms of the contributions of the study, it was not clear if the findings regarding “stakeholder perceptions” were true or not/biased. In other words, these seem like white-collar perspectives about blue collared workers/work. This is a limitation of the study. Kindly clarify these aspects in the Discussion and/or revise accordingly.
Specific Comments
Line 27: The authors write “Subsequently, HS contacted organizations ….” At first it was not clear what was meant by HS. It would be clearer to state this as “Subsequently, the lead author (HS) contacted …” or “the principal research (HS)…” as stated later on line 117. Use direct name or abbreviations of people (e.g., . “Prior to the interviews, HS explained…” (line 120) is not recommended practice. Please review and revise throughout the manuscript.
Table 1: A lot of the information in this table is redundant. Perhaps the authors could consider summarizing this information in 1-2 bar charts. Also, if was unclear if the analysis were stratified by any of these categories (e.g., organization, department, position, or industry) and if there were or weren’t any differences/similarities. Please clarify in the Methods and/or Results accordingly.
Table 2. This table was a bit difficult to follow with the bullet lists and sub-bullet lists inside the cells. The right half of the table could use better formatting for easier reading/interpretation. I also recommend proofreading from grammar and sentence structure.
